

# Soil water depletion patterns in rainfed apple orchards and wheat fields

Lu Zhang[1,2,3,4], Yiquan Wang[3] and Zenghui Sun[1,2,3,4]

[1] Shaanxi Provincial Land Engineering Construction Group Co., Ltd., Institute of Land Engineering and Technology, Xi'an, China
[2] Shaanxi Key Laboratory of Land Consolidation, Xi'an, China
[3] Key Laboratory of Degraded and Unused Land Consolidation Engineering, The Ministry of Natural Resources, Xi'an, China
[4] Shaanxi Provincial Land Consolidation Engineering Technology Research Center, Xi'an, China

## ABSTRACT

Agricultural production in the Weibei rainfed highland, Northwest China, is challenged by severe drought and water shortages. While the land use pattern has shifted gradually from crop production to orchard farming in Weibei, little is known about the influence of fruit industry development on regional water resources and the rationality of planting orchards. Here, we characterized soil water depletion patterns in rainfed orchards and farmlands to evaluate the occurrence of soil desiccation under land use conversion from farmlands to orchards in Weibei. Soil moisture dynamics were monitored in the 0–150 cm soil profiles of different aged Red Fuji apple orchards (young: 7 years, mature: 13 years, old: 22 years) and long-term cultivated winter wheat fields. We measured soil moisture content by oven-drying method in the middle of each month during the growing season of apple trees (March–September 2019). The over-depletion and depletion of soil water were analyzed to evaluate water stress and differential water depletion by distinct vegetation, respectively. The soil desiccation index was used to determine the occurrence of dry soil layers. Water stress was only observed at the 0–70-cm soil depths in the old orchards (mid-June) and farmlands (mid-May–mid-July). Water depletion took place at deeper depths for longer periods in the older orchards than in the younger orchards. Soil desiccation was absent in the young orchards, with mild desiccation at the 0–80-cm soil depths in the mature and old orchards in mid-June. The desiccation intensity was mild at the 0–60-cm soil depths in mid-April–mid-May, intense at the 0–150-cm soil depths in mid-June, and moderate at the 20–150-cm soil depths in mid-July. Results of this study demonstrate the mitigation of water stress and soil desiccation following conversion from wheat fields to apple orchards, which verifies the rationality of planting orchards in the rainfed highland area. Our findings provide strong support for developing a novel model of agro-industrial development, ecological construction, and sustainable economy in the vast arid and semi-arid areas of Northwest China.

Corresponding author
Yiquan Wang, 2918173256@qq.com

## INTRODUCTION

The soil water regime is a key factor influencing regional land use, vegetation planning, and ecological construction. Rainfed highlands are characterized by inadequate moisture conditions and poor irrigation facilities (*Yang & Shao, 2000*; *Huang & Gallich, 2006*; *Wang et al., 2011*; *Zhang, 2019*), in contrast to abundant light and heat resources (*Xu, 2009*; *Wang, Li & Xing, 2015*; *Wang, Joost & Zhang, 2016*). Weibei rainfed highland is a critical agricultural region in Northwest China, where water shortage is the primary factor limiting sustainable crop production. Since China implemented the reform and opening-up policy and adjusted agricultural production structure, the land use pattern in Weibei has shifted from crop production to orchard farming (*Zhao et al., 2012*). Such conversion of land use aims to address the shortcomings of regional resources and take advantage of available natural resources. However, little is known about the influence of fruit industry development on regional water resources and the rationality of planting orchards in Weibei over the past four decades.

A myriad of studies have evaluated soil moisture dynamics based on the physiological responses of crops (*Chen et al., 2007*; *Lü et al., 2009*; *Lü et al., 2012*; *Cheng, Liu & Li, 2014*; *Hu & Si, 2014*). *Yang & Shao (2000)* argued that vegetation not only consumes but also conserves soil water. In the absence of significant changes in natural conditions (*e.g.*, climate, soil), land cover strongly influences the occurrence and severity of soil drought (*Li et al., 2004*; *Mahmood & Hubbard Kenneth, 2005*; *Wang, Shao & Liu, 2010*; *Yang et al., 2012*). It has been suggested that soil water storage capacity in vegetated land is substantially higher than in bare or fallow land (*Zhang & Schilling Keith, 2006*; *Giraldo Mario et al., 2008*; *Wang, Liu & Zhang, 2009*). Additionally, the spatiotemporal distribution and migration patterns of soil water vary under different land cover types (*Jacobs et al., 2004*; *Guber et al., 2008*; *Brocca et al., 2010*; *Biswas & Si, 2011*; *Wang, Shao & Liu, 2012*; *Daniel, 2019*; *Du, Zhang & Li, 2021*). For example, *Yang & Tian (2004)* observed different intensities of desiccation below 3-m soil depths in various forestlands, excluding young apple orchards and shrub stands (*Hippophae rhamnoides* and *Caragana korshinskii*); the desiccation intensity increased with increasing age of forest trees. *Tian et al. (2019)* also reported afforestation-enhanced water supply in the 0–3-m soil profile of forestlands in loess hilly areas, despite the occurrence of dry soil layers. Furthermore, *Bai et al. (2017)* observed distinct spatial and seasonal soil water trends in the Qilian Mountains, which were closely related to different land cover types.

According to *Yang & Shao (2000)*, when evaporation is dominant (equivalent to the rate in bare land after crop harvesting), soil water depletion occurs mainly in the upper soil layers; when transpiration is dominant (equivalent to the rate in vegetated land during crop growing seasons), soil water depletion is concentrated mainly in the soil layers with dense roots. This suggests that the location and intensity of soil desiccation change under different water depletion patterns across growing seasons of various vegetation types. In the case of Weibei region, we hypothesized that soil moisture dynamics and water depletion patterns would respond to the conversion of the major vegetation from annual winter

wheat to perennial fruit trees, which in turn influences the occurrence of soil desiccation over temporal scales.

To test our hypothesis, we monitored the spatiotemporal dynamics of soil moisture content and evaluated soil water depletion in different aged apple orchards and long-term cultivated winter wheat fields in Weibei. The changes of soil moisture distribution in orchards and farmlands were analyzed to depict the spatiotemporal variability in soil water migration. Soil water depletion patterns were characterized to uncover the occurrence time and degree of soil drought, which enabled the quantification of soil desiccation under apple orchards and farmlands. Results of this study are expected to provide empirical evidence for farmers and policy makers to implement rational land use conversion in arid and semi-arid areas of Northwest China.

## MATERIALS AND METHODS

### Site description

The study was conducted in Zhanghong Town, a typical rainfed highland area in Weibei (108°08′–108°52′E, 34°57′–35°33′N), located in the central part of Xunyi County, Shaanxi Province, China (Fig. 1). The study area falls under the fruit breeding development area in the Weibei loessial rainfed highland. The complex terrain has a mean elevation of 1,155 m a.s.l., with higher relief in the northeast and lower relief in the southwest. The landform exhibits typical hilly and gully features of the Loess Plateau. The mean annual temperature is 9.1 °C, being coldest month in January (minimum temperature −24.3 °C) and hottest in July (maximum temperature 36.3 °C). The annual mean temperature difference is −26 °C, and the mean frost-free period is ∼180 days.

Xunyi has abundant sunlight resources, with an annual total solar radiation >500 kJ cm$^{-2}$ and annual total sunshine hours >2,300 h; the sunlight resources are most abundant in May and June. The considerable diurnal and annual temperature differences are favorable for sugar accumulation in apples and high-quality fruit production. In the year of the study (2019), the mean monthly temperature was 6.4 °C in March (sprouting in spring) and 15.4 °C in September (harvesting in autumn); the >0 °C cumulative temperature and ≥10 °C effective cumulative temperature were 3834.5 °C and 3534.1 °C, respectively. The monthly rainfall and evaporation data of 2019 are shown in Fig. 2.

The test soil was a dark loessial soil (Cumuli-UsticIsohumosols). Given the negligible runoff in apple orchards and farmlands with high ridges and flat terrain, the permeable soil was almost able to receive all the natural rainfall, and the depth of rainfall infiltration recharge ranged from 1.6 to 2.0 m or greater. Background soil properties were determined with samples collected from the study area in 2019. The soil pH (8.1–8.4) was measured in a 1:2.5 soil: water suspension using a PHS-3C pH meter (Leici, Shanghai, China). The mean soil bulk density (1.3 g cm$^{-3}$) was determined in the 0–150 cm soil profile using a 100-cm$^3$ cutting ring. Soil particle size composition was determined using an APA2000 Malvern laser particle size analyzer (Malvern, Worcestershire, UK). The soil contained 15.23% sand, 59.14% silt, and 25.63% clay, with a loam texture. Soil organic matter was determined using a 476026 digital bottle-top burette (BrandTech Titrette[®], Essex, CT, USA). Total

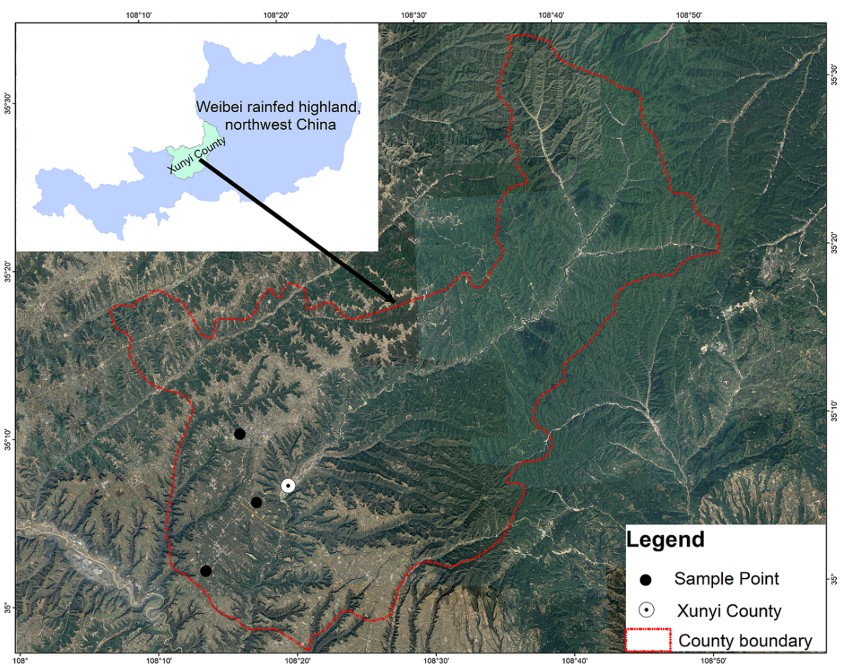

**Figure 1** **Map of the study area in Xunyi County, Shaanxi Province, Northwest China.** (The remote sensing image used in this study is freely available at the Geospatial Data Cloud (http://www.gscloud.cn)).

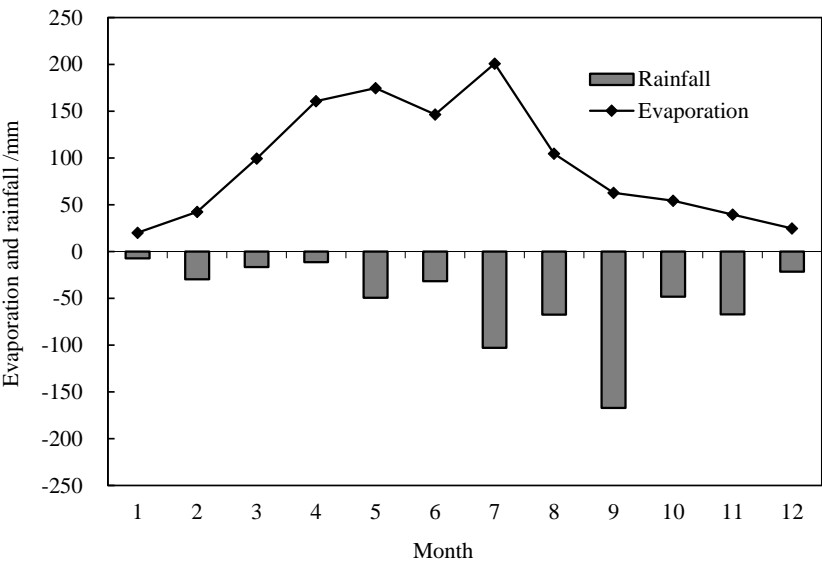

**Figure 2** **Monthly rainfall and evaporation in Xunyi County, Shaanxi Province, China for 2019 (measured using a stainless ombrometer and a cylinder evaporator every day from January to December).**

nitrogen, phosphorus, and potassium were analyzed using a UDK 129 Kjeldahl Destillations System (VELP Scientifica, Deerpark, NY, USA), a TU-1810 UV-visible spectrophotometer (Purkinje, Shanghai, China), and a FP640 flame photometer (Aopu, Beijing, China),

respectively. The topsoil contained: organic matter content, 9.7–11.7 g kg$^{-1}$; total nitrogen, 0.4–1.0 g kg$^{-1}$; total phosphorus, 1.5–1.6 g kg$^{-1}$; and total potassium, 21.2–24.2 g kg$^{-1}$.

## Experimental design and soil sampling

Apple (*Malus domestica* Borkh. cv. 'Red Fuji') orchards of different ages (young: 7 years, mature: 13 years, old: 22 years) were selected in the Xunyi Apple Experiment Station of Northwest A&F University, with three orchards per group. Our access for field sampling was approved by Xunyi Agricultural and Rural Bureau and local farmers who had contracted the orchards. Apple trees were planted 2 m apart with rows 4 m apart in each orchard. Winter wheat (*Triticum aestivum* L. cv. 'Xiaoyan 22') fields 20 m apart were selected in the vicinity of orchards as controls, with one plot per group. The plot sizes of orchards and farmlands were 200,000 m$^2$ (500 m long × 400 m wide).

Soil moisture content was monitored in each plot from the beginning of March in 2019, when apple trees sprouted, to the end of September in 2019, when apple fruits were harvested. During the growing season of apple trees, soil sampling was conducted once in the middle of each month (seven times in total), which avoided rainy days. The sampling points avoided fertilization sites and the sampling intervals were chosen avoid transitional soil horizons: 0–40 cm (loess alluvial deposit with dramatic moisture variations, in which smaller intervals were used for moisture measurements); 40–80 cm (dark loessial soil); and 80–150 cm (loess parent material with minor moisture variations).

During each sampling activity, a 6 cm-diameter soil auger was used to collect soil samples at eight depth intervals in the 0–150 cm profile (0–10, 10–20, 20–40, 40–60, 60–80, 80–100, 100–120, and 120–150 cm) (*Thakur, 2022*; *Thakur et al., 2022a*). In the orchard plots, four sampling points were selected 1 m away from the trunks of five random trees in four directions symmetrically, and the four samples from each tree were mixed to form a composite sample. In the farmland plots, soil samples were obtained at five points based on the diagonal method (one at the central point of the plot and four at the mid-point of the straight lines connecting the central point to the four corners) (Fig. 3). Soil moisture conditions were monitored in the morning (8:00 am–12:00 pm) once a month and additional measurement was made on the day after rainfall.

## Data analysis

Soil water depletion and relevant parameters were calculated based on depth and time (month) using Eqs. (1)–(3).

$$SWO = (MAD - \theta_V) \times H \times 10 \tag{1}$$

$$SWD = TSW_1 - TSW_2 \text{ (where } TSW = \theta_V \times H \times 10) \tag{2}$$

$$SDI = \left(1 - \frac{\theta_V - WM}{MAD - WM}\right) \times 100\% = \frac{MAD - \theta_V}{MAD - WM} \times 100\% \tag{3}$$

In Eq. (1): *SWO* is soil water over-depletion, mm; $\theta_v$ is soil moisture content by volume, %; *MAD* is maximum allowable depletion (arithmetic mean of field capacity (measured
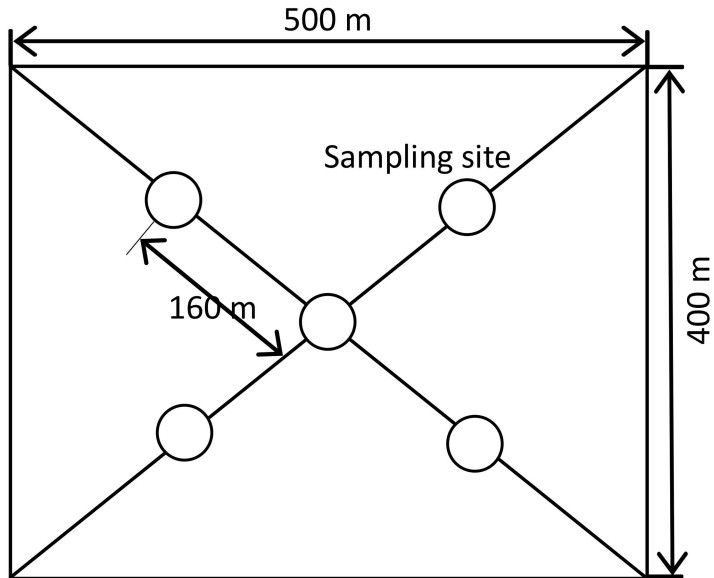

**Figure 3  Distribution of sampling points in each plot data analysis.**

using a 100-cm$^3$ cutting ring and averaged for each soil layer) and wilting coefficient), %; and $H$ is soil depth divided by 10 ($H = 1, 2, 3\ldots$), cm.

In Eq. (2): $SWD$ is soil water depletion, mm; $TSW_1$ is soil water storage at the precedent sampling time $t_1$, mm; and $TSW_2$ is soil water storage at the next sampling time $t_2$, mm.

In Eq. (3): $SDI$ is soil desiccation index, %, where $WM$ is permanent wilting coefficient, (determined with maize seeds in 250-mL aluminum boxes and averaged for each soil layer).

Based on the $SDI$ value, soil desiccation intensity could be divided into the following six levels: (1) $SDI < 0$, no desiccation; (2) $0 \leq SDI < 25\%$, mild desiccation; (3) $25\% \leq SDI < 50\%$, moderate desiccation; (4) $50\% \leq SDI < 75\%$, severe desiccation; (5) $75\% \leq SDI < 100\%$, intense desiccation; and (6) $SDI \geq 100\%$, extreme desiccation (*Cao et al., 2012*).

Data are means $\pm$ standard deviation ($n = 5$). Statistical analyses were performed using the $t$-test in IBM SPSS Statistics 20.0 (IBM Corp., Armonk, NY, USA). A $P$ value less than 0.05 was considered to indicate statistical significance. The error in this study was mainly resulted from the environmental error—the uncertainty and inconsistency of five sampling points selected in each plot.

## RESULTS

### Profile distribution of soil moisture

The distribution patterns of soil moisture in the 0–150-cm profile of orchards and farmlands are illustrated in Fig. 4. Soil moisture content was substantially different between the two land cover types and variably changed among the three age groups of apple orchards. The

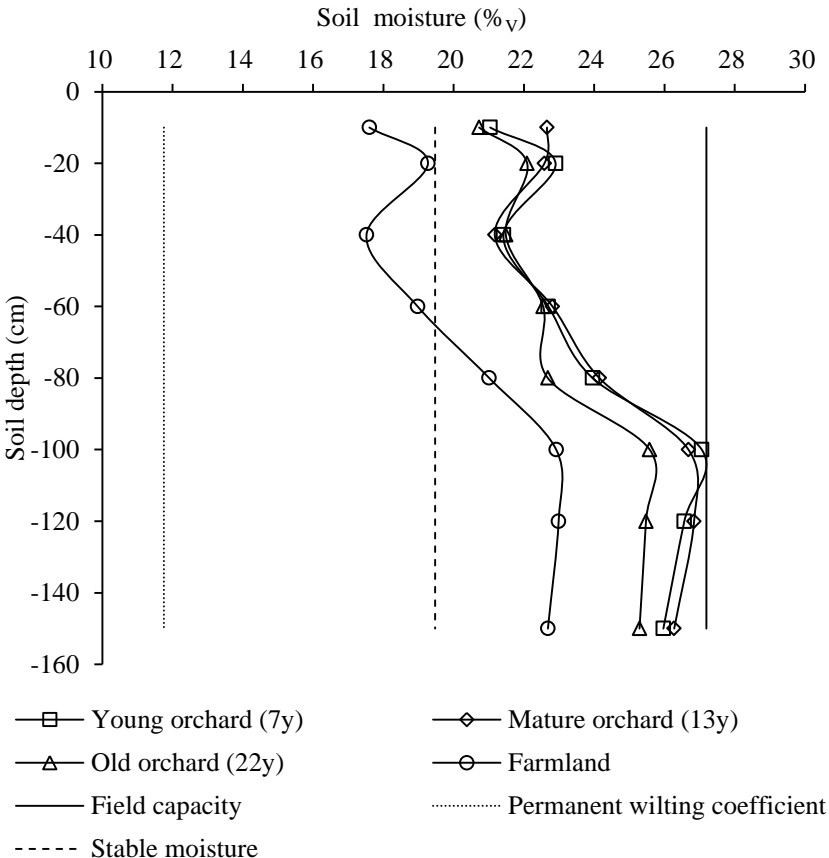

**Figure 4** **Soil moisture distribution in the 0–150-cm profile in apple orchards and farmlands.**

mean soil moisture contents in the profiles of young, mature, and old orchards were 17.6%, 18.5%, and 14.1% higher than those of farmlands, respectively.

In contrast with local stable soil moisture content, the minimum soil moisture contents in the profiles of young, mature, and old orchards were 8.1%, 8.7%, and 6.4% higher, respectively, whereas the moisture content at the 0–60-cm soil depths of farmlands was 5.81% lower. With increasing depth, soil moisture content exhibited increased in farmlands and exceeded the local stable moisture content at the 60-cm depth, then levelling off in deeper soil layers.

### Differences in soil water over-depletion

The estimates of soil water over-depletion were negative values with minor differences between groups in most cases, and positive values were obtained from the old orchards only in mid-June (Fig. 5). During the apple tree-growing season, the mean soil moisture contents in the 0–150-cm profiles of young, mature, and old orchards were 23.0%, 24.0%, and 19.3% higher than the local stable moisture content, respectively.

Positive values of soil water over-depletion were obtained from farmlands in mid-May, mid-June, and mid-July. The highest over-depletion occurred in mid-June, which was

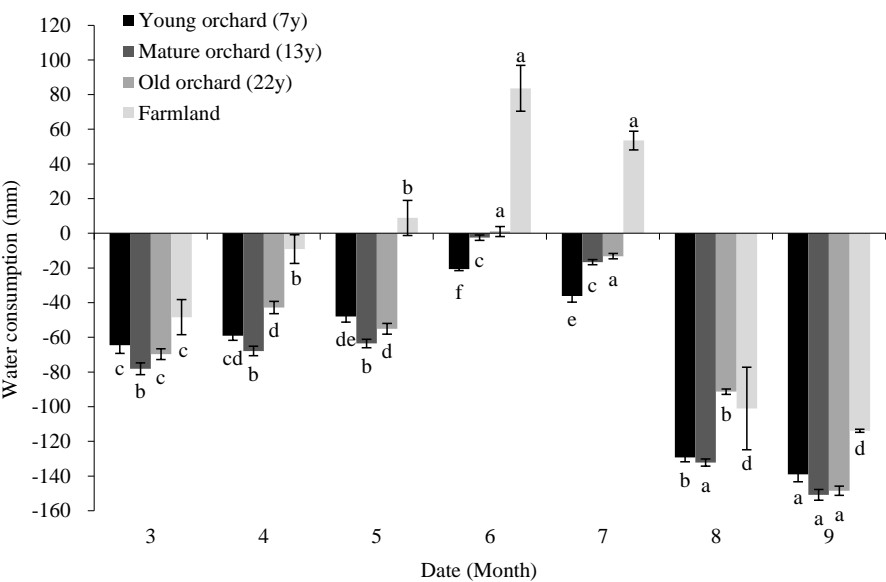

**Figure 5** **Soil moisture depletion in apple orchards and farmlands in the course of an apple-growing season.** Lower case letters above or below the error bars indicate significant differences between groups ($P < 0.05$).

9.4-fold that of mid-May and 1.6-fold that of mid-July. The mean soil moisture content in the farmland soil profiles was 4.9% higher than the local stable moisture content. Additionally, there were significant differences in soil water over-depletion between the orchards and farmlands across different periods ($P < 0.05$).

### Soil water surplus

Soil water depletion ($SWD > 0$) occurred in the 0–150-cm soil profiles of young and mature orchards in mid-May to mid-June. As for the old orchards and farmlands, this phenomenon occurred mainly in mid-March to mid-April, and mid-May to mid-June (Table 1). When compared with orchards, soil water depletion took place at deeper depths for longer periods in farmlands.

Soil water surplus ($SWD < 0$) was observed in the young orchards relative to the mature and old orchards, at 2.7% and 467.9%, respectively. The soil water surplus in the mature orchards and farmlands relative to the old orchards was 453.1% and 398.8%, respectively. From mid-March, there was a surplus of soil water at the 60–80-cm depth in the young orchards, and the depth of soil water surplus changed to the 80–100-cm depth in the mature orchards. In the old orchards and farmlands, a surplus of soil water occurred at the 0–10 cm depth from mid-April.

The total water depletion in the soil profile was ranked as follows: old orchards >farmlands >mature orchards >young orchards. The highest total water depletion was observed from mid-May through mid-June in both orchards and farmlands, before the lowest depletion and highest surplus in mid-July through mid-August. The total water depletion in the mature orchards increased by 122.9% and 8.9% when compared with that

Zhang et al. (2023), *PeerJ*, DOI 10.7717/peerj.15098

**Table 1  Water depletion in apple orchard and farmland soil profiles (mm).**

| Crop | Period | Soil depth (cm) | | | | | | | | |
|---|---|---|---|---|---|---|---|---|---|---|
| | | 0–10 | 10–20 | 20–40 | 40–60 | 60–80 | 80–100 | 100–120 | 120–150 | Σ |
| Young orchards | 3.15–4.15 | 3.16 ± 1.33a† | 0.89 ± 0.37b | 2.55 ± 0.58b | 1.59 ± 0.95a | −0.61 ± 0.79b | −1.12 ± 0.21a | −0.60 ± 0.25a | −0.50 ± 2.10a | 5.36 ± 0.12a |
| | 4.16–5.15 | −6.10 ± 0.62a | −1.15 ± 0.44a | −1.42 ± 1.46a | 2.43 ± 1.17b | 5.74 ± 1.69a | 5.41 ± 1.22a | 4.54 ± 091a | 1.63 ± 0.75a | 11.08 ± 1.76a |
| | 5.16–6.15 | 3.33 ± 1.28a | 2.18 ± 0.08b | 5.91 ± 0.24a | 6.60 ± 0.29a | 1.86 ± 0.23a | 0.28 ± 2.42a | 0.48 ± 2.48b | 6.75 ± 2.21a | 27.39 ± 7.77a |
| | 6.16–7.15 | −7.68 ± 0.16a | −2.68 ± 0.80a | 0.78 ± 1.36a | −1.59 ± 2.74a | −1.40 ± 1.13a | −0.50 ± 1.40a | 0.21 ± 1.84a | −2.61 ± 2.01a | −15.46 ± 0.76a |
| | 7.16–8.15 | −4.80 ± 1.02a | −6.81 ± 0.74ab | −18.05 ± 1.91a | −18.45 ± 4.15a | −15.89 ± 0.18a | −13.39 ± 0.73ab | −12.67 ± 2.78a | −3.18 ± 2.29a | −93.24 ± 2.19a |
| | 8.16–9.15 | 1.70 ± 0.13ab | 1.72 ± 0.36ab | 2.31 ± 0.27b | 1.97 ± 0.67a | 1.86 ± 0.52ab | −1.04 ± 0.36a | −1.19 ± 3.10a | −17.09 ± 2.13a | −9.76 ± 4.39a |
| | Σ | −10.38 ± 0.39b | −5.85 ± 1.08b | −7.92 ± 0.37ab | −7.45 ± 0.80a | −8.44 ± 0.48a | −10.36 ± 0.23a | −9.22 ± 0.37a | −15.00 ± 1.73a | −74.62 ± 0.15a |
| | % | 21.7 | | 10.6 | 10.0 | 11.3 | 13.9 | 12.4 | 20.1 | 100 |
| Mature orchards | 3.15–4.15 | 3.25 ± 0.68b | 3.04 ± 0.42c | 2.80 ± 0.65c | 4.29 ± 0.51a | 1.16 ± 2.69b | −2.56 ± 2.03a | −1.37 ± 1.27a | −0.35 ± 2.17a | 10.25 ± 7.04a |
| | 4.16–5.15 | −3.41 ± 0.67a | −2.98 ± 1.01a | 1.58 ± 1.60a | 3.53 ± 1.10b | 2.45 ± 0.97a | 4.05 ± 1.51a | 3.14 ± 1.00a | −4.05 ± 0.35b | 4.31 ± 0.88ab |
| | 5.16–6.15 | 6.37 ± 1.94a | 6.95 ± 2.11a | 12.78 ± 3.78a | 5.82 ± 3.53a | 5.34 ± 4.46a | 6.50 ± 4.16a | 5.53 ± 3.04ab | 11.76 ± 5.01a | 61.05 ± 20.82a |
| | 6.16–7.15 | −7.36 ± 1.66a | −6.64 ± 1.54ab | −4.58 ± 3.58a | 2.35 ± 4.10a | 1.68 ± 4.28a | 1.68 ± 3.14a | −1.77 ± 3.21a | 0.54 ± 2.19a | −14.10 ± 20.44a |
| | 7.16–8.15 | −5.77 ± 1.18a | −4.97 ± 0.87a | −21.37 ± 2.79a | −27.43 ± 1.46ab | −21.84 ± 0.87a | −17.27 ± 1.14ab | −9.17 ± 2.58a | −7.77 ± 5.41a | −115.60 ± 0.70ab |
| | 8.16–9.15 | 2.16 ± 0.76ab | 1.24 ± 1.16ab | 2.10 ± 0.10b | 3.87 ± 0.58a | 3.69 ± 0.68a | −2.79 ± 0.04a | −8.98 ± 3.30a | −19.86 ± 4.39a | −18.59 ± 9.39a |
| | Σ | −4.77 ± 1.28a | −3.38 ± 0.46a | −6.69 ± 1.59a | −7.56 ± 1.27a | −7.53 ± 2.42a | −10.39 ± 2.02a | −12.63 ± 0.56a | −19.73 ± 0.18a | −72.68 ± 0.59a |
| | % | 11.2 | | 9.2 | 10.4 | 10.4 | 14.3 | 17.4 | 27.2 | 100 |
| Old orchards | 3.15–4.15 | 3.28 ± 1.41a | 4.65 ± 1.15a | 2.87 ± 2.05b | 3.98 ± 1.27a | 5.49 ± 0.70a | 1.40 ± 0.15a | 1.88 ± 0.18a | 3.35 ± 1.60a | 26.88 ± 3.30a |
| | 4.16–5.15 | −3.49 ± 3.20a | −3.26 ± 2.16a | −3.69 ± 1.04a | −0.13 ± 1.00b | 1.86 ± 0.73a | 1.38 ± 1.18a | −2.48 ± 1.89b | −2.41 ± 1.26b | −12.22 ± 1.75b |
| | 5.16–6.15 | 2.08 ± 3.75a | 3.22 ± 1.48ab | 10.38 ± 2.19a | 6.70 ± 0.75a | 4.60 ± 0.72a | 8.88 ± 2.64a | 9.44 ± 3.27ab | 10.77 ± 3.97a | 56.06 ± 8.33a |
| | 6.16–7.15 | −6.96 ± 2.91a | −4.03 ± 0.85ab | −2.86 ± 1.98a | 3.59 ± 0.46a | −0.39 ± 1.52a | −2.06 ± 3.54a | −1.22 ± 3.84a | −0.31 ± 5.54a | −14.24 ± 12.20a |
| | 7.16–8.15 | −5.22 ± 0.38a | −6.04 ± 0.57ab | −20.55 ± 3.07a | −24.42 ± 2.42ab | −14.31 ± 4.42a | −6.05 ± 5.55a | −1.20 ± 2.89a | −0.28 ± 3.36a | −78.09 ± 21.90a |
| | 8.16–9.15 | 0.79 ± 0.85b | 0.47 ± 0.22b | 2.26 ± 0.67b | 2.51 ± 0.98a | −3.44 ± 2.63b | −13.37 ± 2.20b | −17.25 ± 1.57a | −29.19 ± 2.05a | −57.21 ± 9.96a |
| | Σ | −1.59 ± 0.25b | −0.83 ± 0.72ab | −1.93 ± 0.75b | −1.30 ± 0.38a | −1.03 ± 0.99a | −1.64 ± 1.41a | −1.81 ± 0.98a | −3.01 ± 0.09a | −13.14 ± 3.05a |
| | % | 18.4 | | 14.7 | 9.9 | 7.9 | 12.5 | 13.7 | 22.9 | 100 |
| Farmlands | 3.15–4.15 | 6.95 ± 1.34a | 6.00 ± 0.73a | 10.08 ± 0.92a | 4.12 ± 1.02a | 0.25 ± 1.52b | 1.80 ± 2.78a | 4.34 ± 5.15a | 5.66 ± 7.66a | 39.21 ± 18.44a |
| | 4.16–5.15 | −7.36 ± 1.26a | −2.58 ± 0.58a | 3.27 ± 3.57a | 4.52 ± 1.86a | 6.52 ± 2.91a | 5.75 ± 1.59a | 3.28 ± 1.78a | 4.61 ± 1.30a | 18.01 ± 12.44a |
| | 5.16–6.15 | 5.99 ± 0.48a | 6.05 ± 0.73ab | 8.15 ± 2.35a | 8.64 ± 3.11a | 9.40 ± 2.89a | 11.34 ± 3.44a | 11.65 ± 3.84a | 13.54 ± 5.47a | 74.77 ± 21.21a |
| | 6.16–7.15 | −7.73 ± 1.59a | −7.85 ± 1.71b | −7.30 ± 2.54a | 0.05 ± 2.31a | −0.07 ± 1.04a | −1.34 ± 1.80a | −2.94 ± 2.00a | −2.96 ± 3.14a | −30.13 ± 14.51a |
| | 7.16–8.15 | −6.94 ± 0.88a | −9.34 ± 1.53b | −26.49 ± 2.95a | −32.29 ± 2.30b | −30.41 ± 1.00b | −24.49 ± 4.43b | −14.30 ± 7.89a | −10.25 ± 9.69a | −154.52 ± 21.82b |
| | 8.16–9.15 | 4.24 ± 1.20a | 2.74 ± 0.34a | 6.48 ± 0.97a | 3.72 ± 0.98a | 4.40 ± 1.91a | −1.69 ± 5.38a | −9.45 ± 9.10a | −23.32 ± 9.16a | −12.88 ± 23.54a |
| | Σ | −4.84 ± 0.46a | −4.98 ± 0.28ab | −5.80 ± 1.51a | −11.24 ± 2.37a | −9.90 ± 0.24a | −8.63 ± 1.94a | −7.44 ± 3.38a | −12.71 ± 8.51a | −65.54 ± 10.65a |
| | % | 15.0 | | 8.9 | 17.2 | 15.1 | 13.2 | 11.4 | 19.4 | 100 |

**Notes.**
†Lower case letters in the same column indicate significant differences between different groups ($P < 0.05$).

of young and old orchards, respectively, although it decreased by 18.3% relative to that of farmlands.

The vertical distribution of soil water depletion varied based on orchard age. The highest water depletion was observed at the 40–60-cm depth in the young orchards, the 20–40-cm depth in the mature orchards and farmlands, and the 60–80-cm depth in the old orchards. The lowest water depletion was observed at the 0–20-cm depth in the young orchards and the 120–150-cm depth in the other three groups.

## Soil desiccation intensity and dry soil thickness

Table 2 lists the soil desiccation intensity in apple orchards and farmlands, which showed spatiotemporal changes. Considering the mean moisture content in the soil profile at a particular time, soil desiccation did not occur in the young orchards. However, mild soil desiccation was observed in the mature and old orchards in mid-June.

Next, soil desiccation was analyzed at a more accurate scale based on the moisture content of each soil depth. In the young orchards, mildly dry soil layers occurred at the 0–10-cm depth in March, 20–80-cm depths in June, and 20–40-cm depth in July. Moderately dry soil layers occurred at the 0–10-cm and the 40–60-cm depths in June, and in the 40–60-cm depth in July. Severely dry soil layers were observed at the 0–10-cm depth in April.

In the mature orchards, mildly or severely dry layers were not formed in spring. Dry soil thickness decreased in June, and the dry soil layer moved downward to the 60–80-cm depth in July, when compared with the case in the young orchards. The thickness of moderately dry soil layers increased by 10 cm in June and they moved upward to the 20–40-cm depth in July relative to the young orchards. Severely dry soil layers occurred mainly at the 20–40-cm (June) and 40–60-cm (July) depths in summer.

In contrast with the younger orchards, soil desiccation intensity increased in the old orchards. Slightly dry soil layers were observed at the 0–40-cm depths for five months (March–July). The thickness of moderately dry soil layers increased by 10 cm in June, and they moved downward to the 60–80-cm depth in July when compared with those of mature orchards. The occurrence of severely dry soil layers was similar to the case in the young orchard in spring and the mature orchards in summer.

In farmlands, mildly dry soil layers were present at the 0–10-cm and 40–60-cm depths in March, which shifted to the 0–20-cm depths in May and the 100–150-cm depths in July. Moderately dry soil layers occurred across the 20–150-cm depths in April–July. Conversely, severely dry soil layers only occurred at the 10–60-cm depths in April–May and contracted to the 20–40-cm depths in July. Intensely dry soil layers were distributed at the 0–10-cm and 60–80-cm depths in June–July. Extremely dry soil layers appeared at the 0–10-cm depth in March and then expanded to the 10–60-cm depths in June–July.

## DISCUSSION

In the Weibei rainfed highland, agricultural production has shifted from cultivating crops to orchard farming, which inevitably influences soil moisture conditions. The changes in soil moisture content are mainly associated with spatial differences in root water depletion,

plant transpiration, and near-surface evaporation from vegetation canopy (*Liu & Shao, 2015*). Here we present a comparative analysis of soil water depletion in the 0–150 cm profile between rainfed wheat fields and apple orchards after farmland conversion in Weibei. The results provide a holistic picture of water conditions in planted orchards and potential factors mitigating soil desiccation compared with farmlands.

As the canopy of apple trees has increased, there is serious consumption of soil water, and periodic water deficit in the soil becomes a key factor threatening the healthy growth of apple trees. Additionally, deep soil water consumption cannot be balanced due to high evaporation and low rainfall in the study area. The soil water deficit is also partially attributed to biological utilization, especially in the case of irrational agricultural structure and extensive cultivation.

## Occurrence of soil desiccation in orchards and farmlands

A positive value of soil water depletion (*SWD*) indicates a deficit and loss of water, whereas a negative value implies a surplus and gain of water (*Qi, Hu & Song, 2019*). We analyzed the *SWD* values within specific depths (0–150-cm profile) and periods (apple tree-growing season) in orchards and farmlands. We found a substantial increase in *SWD* from March through mid-June, accompanied by a rise in temperature and robust physiological activity in apple trees. After the arrival of the rainy season in mid-June, the soil moisture was replenished. Based on the total water depletion of the soil profile, the highest *SWD* in orchards occurred in mid-May to mid-June. The results indicate that the depletion of soil water in apple orchards was closely linked to the evaporation under canopies as well as bare ground evaporation between apple tree rows.

Soil water over-depletion (*SWO*) is defined as the difference between stable soil moisture content (*i.e.,* water stress point) and forest soil moisture content (*Cao et al., 2012*). The principle is to measure the level of soil water stress on forests based on soil moisture content being below the water stress point. A positive *SWO* value indicates that the trees are under water deficiency stress; the higher the value, the more severe the stress. Our results showed that the actual soil moisture content in orchards was always higher than the local stable soil moisture content. Therefore, there was almost no water stress in orchard soils, and as such, soil desiccation was generally absent in the apple orchards.

*SDI* is a quantitative indicator of soil desiccation. According to relevant evaluation thresholds (*Yang, Zhang & Chen, 2018*; *Gou et al., 2019*), we analyzed spatiotemporal changes in soil desiccation in apple orchards of different ages based on depth and time (month). Dry soil layers were present mainly at shallow depths in the young orchards, most likely a result of persistent soil drought and water depletion by apple trees. The shallow depth of dry soil layers in the young orchards was favorable for replenishment of soil moisture, and there were no deep dry soil layers with difficulty in recovering moisture. Similar results were observed in the mature orchards, where dry soil layers were formed at the shallow depths in close association with the persistent drought and the physiological cycles of apple trees. However, the dry soil layers in the old orchards expanded in both time and space compared with those in the younger orchards. Our result is compatible with the findings of *Zhang et al. (2020)*, and the underlying cause is that older trees could
**Table 2   Soil desiccation intensity and dry soil thickness in orchards and farmlands in Weibei.**

| Crop | Time (month) | SDI (%) | Desiccation intensity | Soil depth (cm) | | | | | Total thickness of dry soil layer (cm) |
|---|---|---|---|---|---|---|---|---|---|
| | | | | Extreme | Intense | Severe | Moderate | Mild | |
| Young orchards | 3 | −46.8 | \ | \ | \ | \ | \ | 0–10 | 10 |
| | 4 | −38.5 | \ | \ | \ | 0–10 | \ | \ | 10 |
| | 5 | −37.1 | \ | \ | \ | \ | \ | \ | 0 |
| | 6 | −11.7 | \ | \ | \ | \ | 0–10, 40–60 | 20–40, 60–80 | 70 |
| | 7 | −32.8 | \ | \ | \ | \ | 40–60 | 20–40 | 40 |
| | 8 | −117.3 | \ | \ | \ | \ | \ | \ | 0 |
| | 9 | −117.0 | \ | \ | \ | \ | \ | \ | 0 |
| Mature orchards | 3 | −64.4 | \ | \ | \ | \ | \ | \ | 0 |
| | 4 | −50.1 | \ | \ | \ | \ | \ | \ | 0 |
| | 5 | −51.4 | \ | \ | \ | \ | \ | 40–60 | 20 |
| | 6 | 6.5 | Mild | \ | \ | 20–40 | 0–20, 40–60 | 60–80 | 80 |
| | 7 | −17.9 | \ | \ | \ | 40–60 | 20–40 | 60–80 | 60 |
| | 8 | −118.4 | \ | \ | \ | \ | \ | \ | 0 |
| | 9 | −124.1 | \ | \ | \ | \ | \ | \ | 0 |
| Old orchards | 3 | −52.5 | \ | \ | \ | \ | \ | 0–10 | 10 |
| | 4 | −24.6 | \ | \ | \ | 0–10 | \ | 10–20 | 20 |
| | 5 | −40.0 | \ | \ | \ | \ | \ | 0–10 | 10 |
| | 6 | 6.6 | Mild | \ | \ | \ | 0–10, 20–80 | 10–20 | 80 |
| | 7 | −14.8 | | \ | \ | 40–60 | 60–80 | 20–40 | 60 |
| | 8 | −88.3 | \ | \ | \ | \ | \ | \ | 0 |
| | 9 | −123.8 | \ | \ | \ | \ | \ | \ | 0 |
| Farmlands | 3 | −0.4 | \ | | | | | 0–10, 40–60 | 30 |
| | 4 | 0.1 | Mild | 0–10 | | 10–20 | 20–60 | | 60 |
| | 5 | 0.1 | Mild | | | 20–60 | 60–80 | 0–20 | 80 |
| | 6 | 0.8 | Intense | 10–60 | 0–10, 60–80 | | 80–150 | | 150 |
| | 7 | 0.4 | Moderate | 40–60 | 60–80 | 20–40 | 80–100 | 100–150 | 130 |
| | 8 | −0.9 | \ | | | | | | 0 |
| | 9 | −0.9 | \ | | | | | | 0 |

utilize deep soil water, leading to relatively severe soil water deficit at deeper depths in old orchards.

Further analyses revealed that the total dry soil thickness reached 60 cm in both the mature and old orchards from mid-June to mid-July, and severe desiccation occurred at specific soil depths. The phenomenon is directly related to the robust physiological activities of apple trees and persistent drought in spring. As dry soil layers appeared at relatively shallow depths for a short period, they might have minimal influence on the growth and development of apple trees. Additionally, the soil profile in farmlands showed five different levels of soil desiccation intensity (2)–(6), indicating the occurrence of dry soil layers across broad spatial ranges and intensities, in addition to long periods.

## Recovery of soil moisture in orchards converted from farmlands

Given its poor soil water conditions, Weibei is generally considered only suitable for the production of herbaceous plants, whereas arbor trees may not be grown sustainably. Additionally, the high transpiration losses *via* forest canopies can facilitate the formation of dry soil layers in this region (*Yang & Tian, 2004*). With regard to the causes of dry soil layers in the Loess Plateau region, it is proposed that when the original soil water conditions reach a certain level of desiccation, continuous tree and shrub planting results in further soil water depletion, which could worsen soil water conditions (*Cao et al., 2012*; *Li et al., 2008*). Consequently, planting fruit trees in Weibei is thought to pose risks with regard to water resource availability.

Notably, we did not observe any permanent dry soil layers in the apple orchards. Despite the dry soil layers formed in particular periods or at specific depths, they could be restored following rainfall. In contrast, different intensities of soil desiccation were observed at the 0–60-cm depths in farmlands, which could not recover moisture easily. Lack of dry soil layers in orchards is partly due to shading by fruit trees, which prevents soil water evaporation losses (*Jutamanee & Onnom, 2016*). Additionally, Weibei has a relatively flat terrain, so the influence of surface and underground runoff on soil water balance is negligible (*Jiang, 1999*). Furthermore, the loessial soil is deep and thick with groundwater levels of 40–80 m, which have no recharge limitations.

In the absence of irrigation, the soil water input in apple orchards is dependent on natural precipitation, whereas the soil water loss is driven by multiple factors, such as tree root absorption, plant transpiration, physiological water depletion, and soil surface evaporation between individual trees. Therefore, planting apple trees does not necessarily lead to the creation of unrecoverable dry soil layers in the rainfed highland area. The fruit industry can develop sustainably in the Weibei area with appropriate orchard management practices.

This study was designed based on the history of agricultural development in Weibei located in the Loess Plateau region. Therefore, our purpose was to clarify which land use type (farmlands represented by wheat *versus* orchards represented by apple) is favorable for regional soil water use. Findings of the comparative study demonstrate the rationality of developing fruit industry in the study area. From the perspective of water resource utilization, Weibei is more suitable for planting apple trees based on the behavioral receptiveness of farmers, as reported by *Thakur et al. (2022b)*.

With regard to the sampling strategy, we collected samples across 0–150 cm soil depths, which covered the depth range of most apple roots (0–40 cm). In fact, the main nutrient-absorbing roots are distributed in 10–40 cm soil depths in the Weibei rainfed highland, given the unstable moisture contents of 0–10 cm soil depth due to water deficit in the surface soil. Taking into account the integrity of soil moisture movement, the sampling depths could meet the water demand of apple trees.

Importantly, *Cheng, Wang & Qi (2021)* found that in the southern Loess Plateau, soil water use depth increased in apple orchards with increasing tree age; thick dry soil layers were formed in orchards converted from farmlands, which attenuated the leading role of piston flow in groundwater recharge in the loess soil area. These findings led us to consider

how soil moisture responds at deeper depths in apple orchards. Furthermore, follow-up studies should pay attention to coordinating the ratio of farmland and orchard areas in order to achieve sustainable soil water use.

## CONCLUSIONS

This study evaluated soil water depletion patterns in rainfed apple orchards and wheat fields in Weibei. We found that the depth range of soil desiccation was narrowed and its duration was shortened in old orchards relative to farmlands. Overall, soil water stress was mitigated in apple orchards, where the dry soil layers could recover. Based on a water depletion perspective, the conversion of wheat fields into apple orchards mitigated soil desiccation. Therefore, results of this study demonstrate the scientificity of planting orchards in Weibei, which provides strong support for developing a new model of agro-industrial development and ecological construction in the vast arid and semi-arid areas of Northwest China. Rational management techniques such as renewal pruning, blossom and fruit thinning, and tree thinning should be adopted to realize sustainable use of regional resources as well as safeguard food security and economic development.

## ACKNOWLEDGEMENTS

The authors acknowledge the anonymous reviewers for their valuable suggestions that helped improve the quality of the manuscript.

### Funding

This work was financially supported by the Fund for Less Developed Regions of the National Natural Science Foundation of China (No. 42167039). The funders had no role in study design, data collection and analysis, decision to publish, or preparation of the manuscript.

### Grant Disclosures

The following grant information was disclosed by the authors:
The National Natural Science Foundation of China: No. 42167039.

### Competing Interests

Lu Zhang and Zenghui Sun are employed by Shaanxi Provincial Land Engineering Construction Group Co. Ltd.

### Author Contributions

- Lu Zhang performed the experiments, analyzed the data, prepared figures and/or tables, and approved the final draft.
- Yiquan Wang conceived and designed the experiments, authored or reviewed drafts of the article, and approved the final draft.
- Zenghui Sun performed the experiments, authored or reviewed drafts of the article, and approved the final draft.
## Data Availability

The data is available in the Supplemental File.

## Supplemental Information

Supplemental information for this article can be found online at http://dx.doi.org/10.7717/peerj.15098#supplemental-information.

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
