# Peer review of "Soil water depletion patterns in rainfed apple orchards and wheat fields"

_PeerJ, doi:10.7717/peerj.15098_

## Round 0.1 · original submission · Minor Revisions

Please revise the manuscript as per comments and suggestions of reviewers.

Reviewer 1 ·

Basic reporting

I have gone through the article titled “Soil water depletion characteristics in rain-fed apple orchards and wheat fields” by Zhang et al. and am of the view that the article is well structured in all aspects. The verbal communication used by authors is good in a way that clarifies their standpoint.

Experimental design

The experimental design seems good as it covers three age groups of Apple orchards along with nearby wheat fields. Moreover the sampling method used for soil is very appropriate for this kind of study from both, Apple orchards and wheat fields. The depth from which soil is taken is enough too for analyzing soil water depletion. However the following two points needs to be addressed from experimental design.

Line 127: Please clarify your view whether about what was approved by “Xunyi Agricultural and Rural Bureau 128 of China”
Line 129: Check the unit of distance for winter wheat fields as it can’t be 20 “cm” apart. Is it the depth where soil dry layer was observed?

Validity of the findings

This article focuses a burning issue of water depletion in soils of Weibei. The results suggest that the water depletion rates in Apple orchards increase with increase in the canopy of trees which poses a serious threat to the water reserves of the soil. This hazard has been reported previously by many researchers. However the authors have mentioned that they didn’t came across the permanent dry layers of soil. Therefore they are of the view that the comparative study suggests the planting of apple orchards with appropriate orchard management practices which may mitigate the soil desiccation in Weibei.

Additional comments

I find this article appropriate for publication in this esteemed journal after clarifying the above mentioned points.

·

Basic reporting

Overall the hypothesis is very well experimented

Experimental design

Mention the time and dates when experiment was conducted

Validity of the findings

Ok

Additional comments

Overall the paper can be accepted after few changes

·

Basic reporting

The manuscript clearly highlights the mission and vision. The sentences are framed accurately and do not need any grammatical enhancement. In addition upgraded literature has been provided to support the findings.

Experimental design

The objectives and research questions are well designed with appropriate use of standard methodology and results.

Validity of the findings

The results are accurately validated and statistically analysed.

Additional comments

Please add the given references to your manuscript text as well in the reference list

1. Physicochemical Analysis and Studies on Soil Profile of Prunus Armeniaca Collected from Mid-Hill of Himachal Pradesh, N Thakur - National Academy Science Letters, 2022
2. Analysis and Extraction of Curcumin at Mid and Late Phase Harvested Curcuma Longa Samples Collected from Western Himalayan Regions, N Thakur, P Sharma, PP Govender, SK Shukla - Chemistry Africa, 2022
3. Drivers for the behavioral receptiveness and non-receptiveness of farmers towards organic cultivation system, N Thakur, M Nigam, R Tewary, K Rajvanshi, M Kumar… - Journal of King Saud University-Science, 2022

Reviewer 4 ·

Basic reporting

The manuscript entitled “Soil water depletion characteristics in rain-fed apple orchards and wheat fields” seems to be very interesting research article in the field of agriculture, horticulture and all allied field. It may attract the attentions of readers, researchers, and other workers in the relevant fields. It needs minor revisions before publication in this journal because journal is committed to publish the well oriented publications. Some of the flaws are indicated below

Experimental design

2. Proper and Specific aims and objectives are missing in the Abstract. Abstract may be improved by addition of aims and objectives. Future aspects, should be added in the abstract.
3. Brief Methodology in the Abstract should be added.
4. Aims and objectives are not clear. What were specific objectives of the present research?? It should be added in the end of introduction. These should include all parameters included in your results and it should state importance in the terms of future research.

Validity of the findings

5. Results are found to be confusing. Results should be presented in simple format.
6. Discussion needs comparative analysis with recently published work.
7. Conclusion need improvement. What was concluded in the present work? Future aspects and recommendations should be added.

Additional comments

1. Language of manuscript needs improvement.

---

## Round 0.2 · accepted · Accept

Authors have revised the manuscript as per suggestions. Therefore, I recommend that manuscript can be accepted for publication.

Reviewer 1 ·

Basic reporting

I have gone through the article which is now titled as "Soil water depletion patterns in rainfed apple orchards and wheat fields" and am of the view that it has been significantly improved by authors. Now therefore all of the suggested changes have been incorporated, I find this article fit for publication in PeerJ Life & Environment in its current form.

Experimental design

I find this fit for publication with its original experimental design.

Validity of the findings

As is was stated that the agricultural production in Weibei rainfed highland, was challenged by severe drought and water shortages, however this study proved the rationality of planting orchards in the rainfed highland area as it showed mitigation of water stress and soil desiccation following conversion from wheat fields to apple orchards. The results of this study are linked with research questions and will be encouraging to the local community of Northwest China for shifting to orchard farming.

·

Basic reporting

The manuscript has been revised and upgraded accordingly with clarity in sentence framing.

Experimental design

The design has been incorporated and presented accordingly.

Validity of the findings

Results have been justified providing with the lastest and upgraded discussion

Additional comments

The manuscript has been upgraded and revised fruitfully